# Does early intervention improve outcomes in the physiotherapy management of lumbar radicular syndrome? Results of the POLAR pilot randomised controlled trial

Michael Reddington,[1] Stephen J Walters,[2] Judith Cohen,[3] Susan K Baxter,[4] Ashley Cole[5]

[1]Therapy Services Outpatient Department, Sheffield Teaching Hospitals NHS Foundation Trust, Northern General Hospital, Sheffield, UK
[2]Medical Statistics Group, ScHARR, University of Sheffield, Sheffield, UK
[3]Hull Health Trials Unit, Hull York Medical School, University of Hull, Hull, UK
[4]Section of Public Health, ScHARR, University of Sheffield, Sheffield, UK
[5]Orthopaedic Department, Sheffield Teaching Hospitals NHS Foundation Trust, Northern General Hospital, Sheffield, UK

**Correspondence to**
Michael Reddington;
mreddington1@sheffield.ac.uk

## ABSTRACT

**Objective** To investigate the feasibility of undertaking a definitive randomised controlled trial (RCT).

**Setting** This was a pilot, pragmatic superiority RCT with a qualitative element, recruiting from 14 general practitioner (GP) practices in England.

**Participants** Patients over 18 years of age presenting to their GP with unilateral lumbar radicular syndrome (LRS), defined as radicular pain and/or neurological symptoms originating from lumbar nerve roots, were eligible to participate in the study, those who did not have a clear understanding of the English language or had comorbidities preventing rehabilitation were ineligible.

**Interventions** Participants were randomised into early intervention physiotherapy or usual care with the former receiving their treatment within 2 weeks after randomisation and the latter 6 weeks postrandomisation. Both groups received a patient-centred, goal-orientated physiotherapy programme specific to their needs. Participants received up to six treatment sessions over an 8-week period.

**Outcome measures** Process outcomes to determine the feasibility of the study and an exploratory analysis of patient-reported outcomes, including self-rated disability, pain and general health, these were collected at baseline, 6, 12 and 26 weeks postrandomisation.

**Results** 80 participants were recruited in 10 GP practices over 34 weeks and randomised to (early intervention physiotherapy n=42, usual care n=38). Follow-up rates at 26 weeks were 32 (84%) in the usual care and 36 (86%) in the early intervention physiotherapy group. The mean area under the curve (larger values indicating more disability) for the Oswestry Disability Index over the 26 weeks was 16.6 (SD 11.4) in the usual care group and 16.0 (SD 14.0) in the intervention group. A difference of −0.6 (95% CI −0.68 to 5.6) in favour of the intervention group.

**Conclusions** The results of the study suggest a full RCT is feasible and will provide evidence as to the optimal timing of physiotherapy for patients with LRS.

**Trial registration number** NCT02618278, ISRCTN25018352.

## Strengths and limitations of this study

► This pilot randomised controlled trial (RCT) was conducted in the primary care setting with clinical staff delivering the intervention.
► All feasibility objectives were met, including recruitment and participant attrition, and so the study can directly inform the design and conduct of a definitive RCT.
► Participants self-referred into the study after an introduction from their general practitioner (a prerequisite for ethics approval) and so this group of patients may not be representative of a wider population.
► The diagnosis of lumbar radicular syndrome was made from the clinical history and participant symptomatology and as such it is likely that there was a degree of diagnostic heterogeneity within the study sample.
► This was a pilot RCT and as such all analyses are exploratory.

## INTRODUCTION

Lumbar radicular syndrome (LRS) is a painful and disabling condition, usually of benign causation and in around 90% of cases associated with an intervertebral disc (IVD) prolapse.[1] Symptomatic presentation of LRS is heterogeneous, it can be self-limiting, lasting only a short time with no significant sequelae or can be a major cause of prolonged disability, work loss and long-term healthcare usage with associated costs.[2 3] Lifetime prevalence of LRS is estimated to be between 1% and 43%[4] with an annual incidence of between 1% and 5%.[5]

Around 75% of LRS sufferers will have symptom resolution by 12 weeks, alongside spontaneous resorption of the IVD.[6] However, there is no reliable predictor of early, late or no recovery at all.[7] Treatment guidelines

encourage initial conservative management before considering surgery. Physiotherapy for LRS is commonly employed in the UK for the management of LRS, however, there is a lack of consensus on the type, duration and timing of the physiotherapy intervention.[8] Early intervention physiotherapy for low back pain (LBP) has been found to improve patient outcomes, satisfaction and have lower healthcare usage and associated costs.[9–11] Delayed initiation of physiotherapy has been found to increase healthcare consumption in patients with LRS.[12] This suggests early treatment is important in terms of cost savings and prevention of chronic symptom development[13] as increased symptom duration leads to worse outcomes for patients who undertake both conservative or surgical care.[14 15] Surgery for patients with LRS has been advocated, with optimum timing being between 4 weeks and 6 months after symptom onset.[16 17] Superiority studies of surgery and conservative management show a quicker improvement of patient symptoms in surgical groups, with results at a year showing no significant differences.[18 19] A significant number of patients never have any substantial relief from surgery with unsatisfactory outcomes in over 20% of patients at 5 years.[20 21] The timing of physiotherapy engagement for LRS has yet to be investigated.

## AIMS AND OBJECTIVES

The study aim was to investigate the feasibility of undertaking a full randomised controlled trial (RCT) to determine the effectiveness and cost-effectiveness of early intervention physiotherapy for patients with LRS.

### Process objectives

1. Successfully set-up recruitment sites in general practitioner (GP) practices.
2. Achieve a recruitment rate of seven participants per month.
3. Demonstrate the ability to organise 75% of physiotherapy appointments within 2 weeks of randomisation.
4. Provide an appointment within 20 days of randomisation for >75% of participants randomised to the intervention group.
5. Achieve a participant attendance at >66% of physiotherapy appointments.
6. Achieve a participant attrition rate of <25% over the course of the study.
7. Achieve 80% return of patient-reported outcome measures (PROMS) at 6/52 follow-up.

### Research objectives

1. To test the feasibility, practicality, safety and acceptability of the study design and protocol.
2. Demonstrate acceptability of the primary and secondary outcome measures to patients and clinicians.
3. To inform the sample size calculation for the definitive RCT.

## METHODS

### Design and setting

This was a mixed-methods study comprising an external pilot RCT with an embedded qualitative component in the form of stakeholder interviews in 14 GP practices in a large city in England. Known as the PhysiOtherapy management of LumbAr Radicular syndrome (POLAR) study, the pilot RCT will be presented in this paper. A change was made to the inclusion criteria after 1 week of recruitment, the upper age limit of 70 was removed as this excluded a number of potential participants. The protocol for the study has been published, including extensive details of methods.[22]

### Patient and public involvement

The research question was informed directly from patient feedback on physiotherapy services. Current and past patients who have experienced LRS were involved from the inception to the end of the study in various ways. First, they were involved in developing the research question, iteration of the intervention and the study processes. They were invaluable in developing patient information and insight into recruitment strategies. Finally, they were actively involved in the interpretation of the results and discussions of the next stage of the study. Results will be distributed by email or post to participants who opted to receive the results at consent.

### Randomisation

Information from the baseline dataset was used to randomise the participants using a web-based system. The Oswestry Disability Index 2.0 (ODI)[23] was used as the stratification factor with three levels based on ODI severity[24]: 'mild and moderate' (≤22%–40%), 'severe' (>40%–60%) and 'crippled' (>60%–80%). A blinded block size was used to minimise predictability. The random allocation sequence and block size, stratified by centre and ODI disability score, was independently generated by the Sheffield Clinical Trials Research Unit .

Participants were informed of their group allocation within one working day of their consent and randomisation. Participants were randomised to treatment at either 2 or 6 weeks postrandomisation, we were unable to blind either patients or clinicians to the treatment allocation as it was obvious at what time point they were receiving treatment. In an effort to minimise bias, both groups of patients received protocolised treatment based on the same assessment and treatment framework at the different time points.

### Participants

Potential participants with a clinical diagnosis of LRS were identified by their GP and given details of the study. Each participating GP underwent training and was equipped with a diagnostic aide memoire for clinically identifying patients with LRS (see online supplementary file 1). If interested, the patient contacted a member of the research team who screened for eligibility and arranged

to meet to discuss the study. Anyone over the age of 18 years with unilateral LRS and who could speak English were eligible. If they had 'red flag' signs or symptoms such as cancer, cauda equina syndrome, spinal fracture or had other physical or psychological disabilities preventing rehabilitation, they were ineligible.

### Recruitment and consent

Written consent was obtained by the research team after meeting the potential participant and confirming eligibility criteria including the clinical diagnosis of LRS. There were three recruitment cycles, each lasting up to 12 weeks or until 27 participants had been recruited for that cycle (26 for the final cycle). The remaining 8 weeks were used for completion of treatment. A 2-week period between cycles provided time to reflect and analyse the results from the stakeholder interviews and other feedback to refine the study processes as necessary.

### The intervention

The intervention was protocolised and allowed the treating physiotherapist a range of treatment options within each domain. Selected options were recorded electronically for each treatment session. The goal-orientated physiotherapy regimen for both groups were tailored to the individuals' requirements based on the information gathered from the baseline interview data, PROMS and clinical assessment. Participants were assessed using a multidimensional approach based on seven different elements: psychological barriers to recovery, neurological factors, movement restriction, understanding, conditioning, movement control and pain. Individualised physiotherapy for LBP and LRS is known to be superior and more cost-effective than advice alone,[25 26] it is flexible and directly relevant to the individual and their changing needs. Participants received a maximum of six sessions of

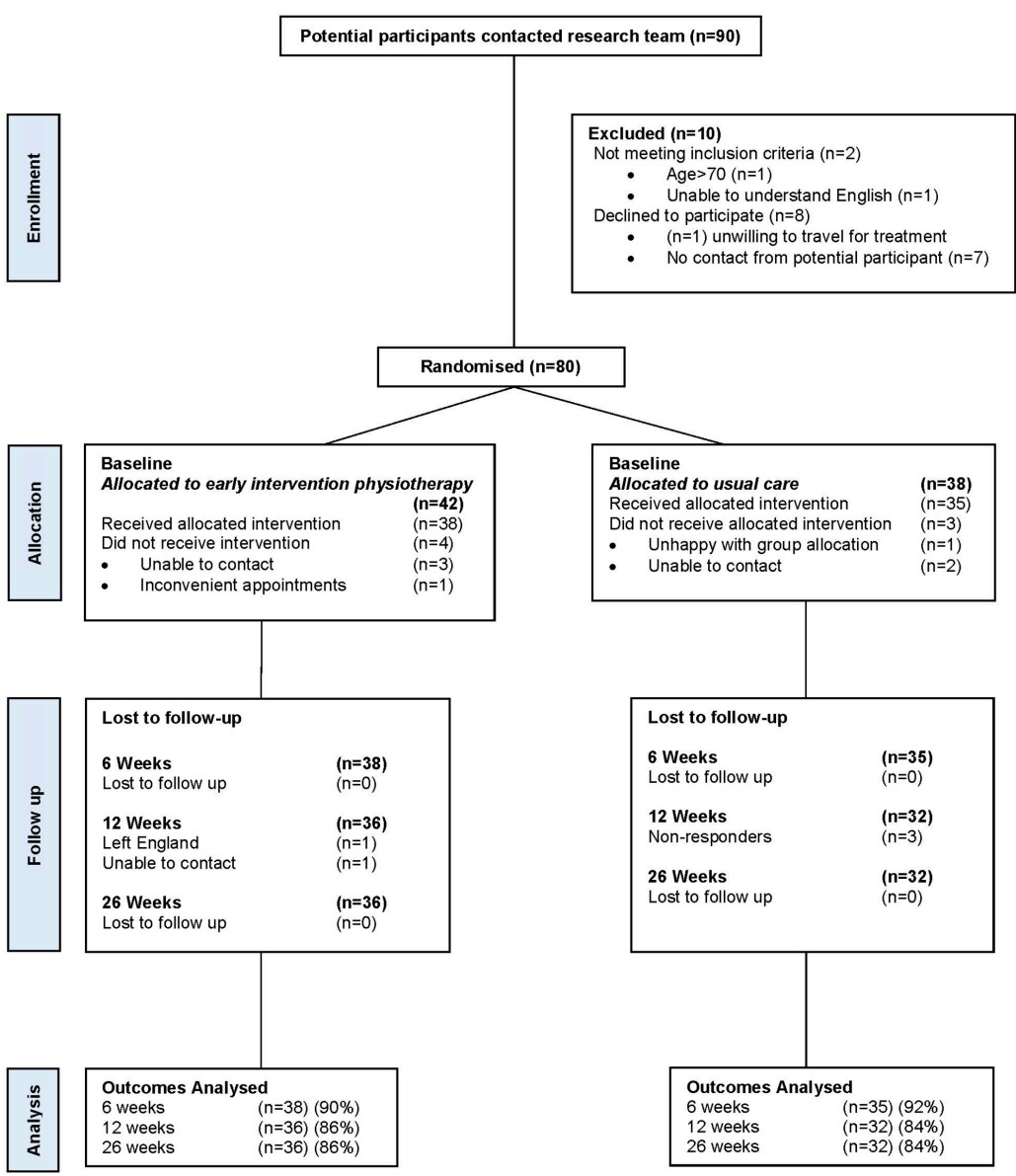

**Figure 1** POLAR Consolidated Standards of Reporting Trials flow chart.

**Table 1** Baseline characteristics of POLAR participants

| | Early intervention physiotherapy | | | Usual care | | | Total | | |
|---|---|---|---|---|---|---|---|---|---|
| | **N** | **%** | | **N** | **%** | | **N** | **%** | |
| Female | 21 | 50 | | 18 | 47 | | 39 | 49 | |
| White British | 38 | 90 | | 33 | 87 | | 71 | 89 | |
| | **N** | **Mean** | **SD** | **N** | **Mean** | **SD** | **N** | **Mean** | **SD** |
| Age (years) | 42 | 47 | 14 | 38 | 47 | 13 | 80 | 47 | 13 |
| Height (cm) | 42 | 172.1 | 10.7 | 38 | 172.1 | 9.8 | 80 | 171.7 | 10.2 |
| Weight (kg) | 39* | 81.5 | 14.8 | 38 | 80.6 | 15.7 | 77 | 81 | 15.2 |
| BMI | 39* | 27.7 | 4.6 | 38 | 27.3 | 5.6 | 77 | 27.5 | 5.1 |
| ODI score (%) | 42 | 44.6 | 19.5 | 38 | 45.2 | 17.4 | 80 | 44.9 | 18.4 |
| Leg pain | 42 | 7.2 | 1.8 | 38 | 6.9 | 2.3 | 80 | 7 | 2.1 |
| Back pain | 42 | 5.4 | 3.3 | 38 | 6 | 2.6 | 80 | 5.7 | 3.0 |
| EQ-5D-5L VAS | 42 | 63.8 | 20.6 | 38 | 64.6 | 18.9 | 80 | 64.1 | 19.7 |
| EQ-5D-5L utility score | 42 | 0.44 | 0.29 | 38 | 0.52 | 0.25 | 80 | 0.48 | 0.27 |
| Keele STarT Back | 42 | 5.7 | 2.0 | 38 | 5.7 | 1.8 | 80 | 5.7 | 1.9 |
| Keele STarT Back subscore | 42 | 2.0 | 1.5 | 38 | 2.7 | 1.3 | 80 | 2.8 | 1.4 |
| Time to treatment (days)† | 38 | 11.1 | 10.5 | 31 | 43.6 | 8.9 | 69 | 25.7 | 19.0 |
| | **N** | **Median** | **IQR** | **N** | **Median** | **IQR** | **N** | **Median** | **IQR** |
| Symptoms duration (days) | 42 | 92 | 276 | 38 | 61 | 51 | 80 | 77 | 203 |

*Three missing values.
†Time between randomisation and first scheduled treatment session.
BMI, body mass index; ODI, Oswestry Disability Index; VAS, Visual Analogue Scale, EQ-5D-5L, EuroQol-5 Dimensions.

physiotherapy over an 8-week period, fewer if their predetermined goals had been achieved. A logic model has been developed for the intervention which can be found in online supplementary file 2.

### Treatment fidelity

Several strategies were employed to optimise fidelity, including a protocolised training package for the treating physiotherapists, standardised patient information, weekly feedback and support of treating physiotherapists and video analysis of each participating physiotherapist treating a study participant. The study took place in an National Health Service (NHS) community setting using three physiotherapists, already employed by the host service provider. The physiotherapists had a mean age of 36 years (range 34–40 years) and a mean of 10 years postgraduate experience (range 7–12 years). They underwent 21 hours of training in the assessment and intervention and to promote and facilitate self-management, optimal

function, pacing advice, analgesic advice together with equipping the patient with coping strategies.

### Outcomes

Patients were asked to complete self-report and screening measures by post or face to face at four-time points: first, at the time of consent and then at 6, 12 and 26 weeks postrandomisation. The primary outcomes for the study were process outcomes as the objective was to determine the feasibility of carrying out a full-scale RCT. Secondary outcomes were the ODI, Visual Analogue Scale for back and leg pain, Keele STarT Back score,[27] EQ-5D-5L[28] and a self-report form focussing on functional loss, goals and medical history.

### Sample size

It has been recommended that an external pilot study should have at least 70 measured participants (35 per group) when estimating the SD for a continuous

outcome.[29] A sample size of 80 patients, with approximately 10% allowance for loss to follow-up allows the SD of an outcome to be estimated to within a precision of approximately ±16% of its true underlying value with 95% CI.

## RESULTS

The flow chart of the participant journey for the POLAR study can be viewed in figure 1. Ninety potential participants who were given details of the study by their respective GPs contacted the research team. Ten were excluded as they either did not meet the inclusion criteria or refused to be randomised, with 80 going on to be randomised from 10 different primary care GP practices.

### Baseline characteristics

The baseline characteristics of all participants, by group can be found in table 1. This illustrates the comparability of the two arms with no evidence of selection bias. The groups were well matched for demographic factors such as age, gender and body mass index as well as levels of disability, pain in leg and back, risk of chronicity and general health status. However, there was evidence of a difference in the EQ-5D utility scores which is attributable to chance as all participants were randomised. The early intervention physiotherapy group had longer symptom duration going into the study.

### Process results

The POLAR study is a pilot trial and outlined below are the results of the feasibility objectives.

### Set-up of recruitment sites in primary care

Twenty GP practices were initially approached to take part in the study, with 10 agreeing to participate. Towards the end of the second tranche of recruitment, it was evident that one practice was recruiting a large number of participants and a decision was made to enrol new recruitment centres. Seven further GP practices were, therefore, approached, with four agreeing to participate.

### Recruitment rate

Eighty participants were recruited between the period 1 March 2016 and 7 November 2016 with a recruitment rate of 2.4 participants per week or 9.6 participants per month which enabled recruitment to end earlier than anticipated. Forty-two participants were randomised into the early intervention group and 38 in the usual care group.

### Organisation of physiotherapy appointments

The target of 75% of physiotherapy appointments being made within 2 weeks of randomisation was surpassed in both groups. One hundred per cent (42/42) (95% CI 92% to 100%) of early intervention physiotherapy participants received their appointment within 20 days of randomisation and 38/38 (95% CI 91% to 100%) in the usual care group. This illustrates the feasibility of making appointments for participants at short notice.

### The feasibility of intervention delivery

A key feasibility parameter was the ability for at least 75% of early intervention physiotherapy participants to be seen by a physiotherapist, within 20 days of randomisation. One hundred per cent (42/42) (95% CI 92% to 100%) of participants reached this target, with a mean of 14.1 days between randomisation and first treatment session.

### Participant treatment session attendance

The mean attendance rate for physiotherapy appointments in both groups was 92.6% (SD 16.2), 93.8% (SD 12.6) for the intervention group physiotherapy and 91.1% (SD 19.8) in the usual care group. All surpassed the a priori target of greater than 66% attendance. The mean number of treatment sessions received by the intervention group was 4 (SD=1) and 3 in the usual care group (SD=2).

### Participant attrition

Eighty participants agreed to take part in the study. The intervention group attrition rate was 14% (6/42) (95% CI 7% to 28%) and in the usual care group it was 16% (6/38) (95% CI 7% to 30%) at 26 weeks follow-up. The overall attrition rate for drop-out of participants was 15% (95% CI 9% to 24%), all within the a priori limit set at 25%.

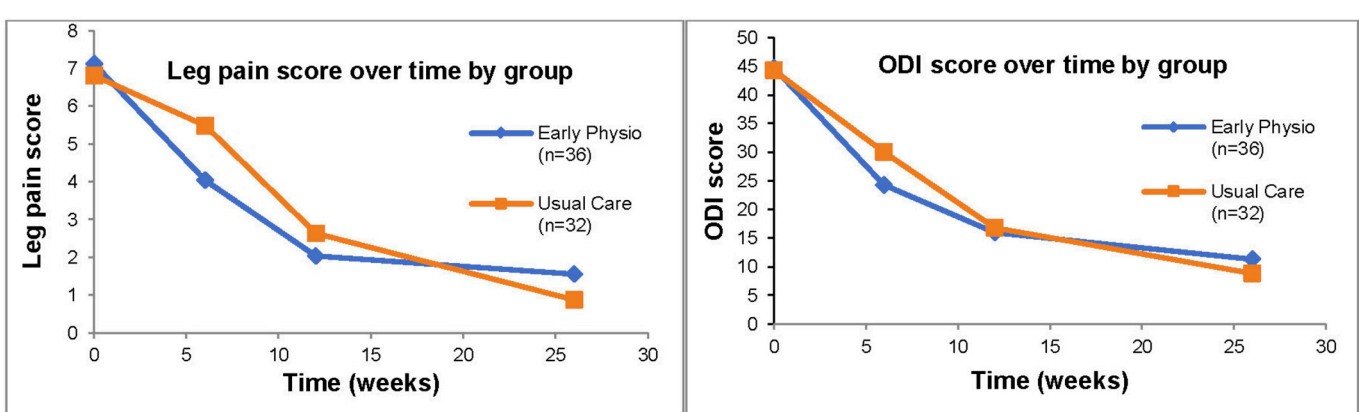

**Figure 2** Leg pain and ODI scores across groups. ODI, Oswestry Disability Index.

## Outcome measure return

The outcome measure return rates surpassed expectations of 80% at 6 weeks and were as follows: 38/42 (91%; 95% CI 78% to 96%) at 6 weeks postrandomisation for the intervention group and 35/38 (92%; 95% CI 79% to 97%) for the usual care group.

## Research results

### Analysis of key clinical outcomes

Figure 2 shows the leg pain and ODI scores (likely primary outcome measures for definitive RCT) for participants with all four assessments completed. The blue line illustrates the increased rate of recovery in the early intervention physiotherapy group up to 6 weeks. When the usual care group begins their physiotherapy, the rate of recovery assimilates and by 12 weeks both groups have very similar scores. The descriptive statistics for all participants by group and time point can be found in table 2. Two participants underwent lumbar microdiscectomy surgery for their LRS. Both participants had completed their respective courses of physiotherapy before undergoing surgery. S05/005 (usual care) failed to make significant improvements to their pain and with a severe level of pain and disability, surgery was undertaken. S06/027 (early intervention physiotherapy) had made significant improvements with physiotherapy, improving by over 20 points on the ODI, but required surgery due to 'impending' cauda equina syndrome.

### The feasibility, practicality, safety and acceptability of the study design and protocol

The feasibility of the study has been suggested by the results of the feasibility parameters. There were several adjustments made to the processes of the study which were made possible by the breaks in recruitment. These included a brief weekly email to all participating GPs to remind them of the study and improve the clarity of inclusion and exclusion criteria. A change to the process of administering the 6-week outcome measures was necessary, after the physiotherapists reported it too time consuming to administer. There were no changes made to the intervention, which appeared to be well received by both participants and clinicians alike. There were no adverse events or serious adverse events (SAEs) associated with the intervention or the study processes.

### Harms

There was one SAE during the course of the study in the early intervention physiotherapy group. The SAE rate was 2% (1/42) in the early intervention physiotherapy group and 0% (0/38) in the usual care group a difference of 2% (95% CI −7% to 12%). The participant was hospitalised after suffering a cerebrovascular accident related to pre-existing vascular hypertension. The participant had completed their physiotherapy intervention 2 weeks prior and made a full recovery at 6 months. This was reported to the ethics committee and Trial Management Group.

**Table 2** Descriptive statistics for outcome measures at each time point

| Outcome | Baseline Usual care n=38 | Baseline Early intervention physiotherapy n=42 | 6 weeks Usual care n=35 | 6 weeks Early intervention physiotherapy n=38 | 12 weeks Usual Care n=32 | 12 weeks Early intervention physiotherapy n=36 | 26 weeks Usual Care n=32 | 26 weeks Early intervention physiotherapy n=36 | AUC Usual care n=32 | AUC Early intervention physiotherapy n=36 | Difference 95% CI Mean | Lower | Upper |
|---|---|---|---|---|---|---|---|---|---|---|---|---|---|
| ODI* (SD) | 45.2 (17.4) | 44.6 (19.5) | 29.1 (16.1) | 24.0 (18.7) | 16.8 (19.2) | 16.0 (19.0) | 8.8 (11.3) | 11.3 (15.5) | 16.6 (11.4) | 16.0 (14.0) | −0.6 | −6.8 | 5.6 |
| VAS back† (SD) | 6.0 (2.6) | 5.4 (3.3) | 4.6 (2.7) | 3.7 (2.6) | 3.1 (2.5) | 2.6 (2.5) | 2.1 (2.1) | 2.7 (2.2) | 1.8 (0.8) | 1.5 (1.0) | −0.3 | −0.7 | 0.1 |
| VAS leg† (SD) | 6.9 (2.3) | 7.2 (1.8) | 5.2 (2.9) | 4.1 (3.0) | 2.6 (2.9) | 2.0 (2.5) | 0.9 (2.2) | 1.6 (2.2) | 1.7 (0.9) | 1.5 (1.0) | −0.2 | −0.6 | 0.3 |
| EQ-5D5L‡VAS (SD) | 64.6 (18.9) | 63.8 (20.6) | 68.9 (16.4) | 72.7 (17.7) | 73.2 (22.9) | 79.6 (17.5) | 81.7 (12) | 79.6 (16.3) | 36.8 (7.1) | 38.1 (7.8) | 1.4 | −2.2 | 5.0 |
| EQ-5D-5L§ utility score (SD) | 0.52 (0.25) | 0.44 (0.29) | 0.7 (0.26) | 0.74 (0.22) | 0.83 (0.23) | 0.85 (0.22) | 0.92 (0.12) | 0.86 (0.19) | 0.39 (0.09) | 0.39 (0.10) | 0.00 | −0.05 | 0.04 |

*Oswestry Disability Index (ODI) 0–100, higher score=higher level of self-rated disability. For the ODI, a larger AUC represents a greater level of disability over the 26 weeks. A negative difference means the early intervention physiotherapy group has the better outcome (lower levels of disability) over the 26 weeks follow-up.
†VAS 0–10, higher score=higher self-report pain. For the VAS back pain and leg pain outcomes, a larger AUC represents a higher level of pain over the 26 weeks. A negative difference means the early intervention physiotherapy group has the better outcome (lower levels of pain) over the 26 weeks follow-up.
‡EQ-5D-5L VAS score, 0–100, self-rated health, the higher the score, the better the quality of life. For the EQ-5D-5L VAS score, a larger AUC represents a higher level of quality of life over the 26 weeks. A positive difference means the early intervention physiotherapy group has the better outcome (higher levels of quality of life) over the 26 weeks follow-up.
§EQ-5D-5L utility score, −0.6 to 1.00 with a higher score representing better quality of life. For the EQ-5D-5L utility score, a larger AUC represents a higher level of quality of life over the 26 weeks. A positive difference means the early intervention physiotherapy group has the better outcome (higher levels of quality of life) over the 26 weeks follow-up.
VAS, Visual Analogue Scale.

**Table 3** Intervention domains and components frequency table

| Domain | No of participants receiving component n=69 | Method of assessment | Treatment options | Frequency of component used | % |
|---|---|---|---|---|---|
| Psychological barriers to recovery[32–34] | 47 (68%) | Keele STarT Back clinical interview and history | 1. Treatment of kinesiophobia with graded exposure, education and movement re-education | 16 | 1.3 |
| | | | 2. Treatment of hypervigilance with education, distraction and desensitisation | 17 | 1.4 |
| | | | 3. Treatment of faulty beliefs about pain, LRS, treatment and/or prognosis with education and self-management strategies | 38 | 3.2 |
| | | | 4. Treatment of iatrogenic beliefs and corresponding avoidance behaviours with education and movement re-education | 3 | 0.2 |
| | | | 5. Treatment of aspects of work as a barrier to recovery and treatment with ergonomic advice and practice | 15 | 1.2 |
| | | | 6. Identification of financial barriers to recovery and signposting, for example, debt management | 15 | 1.2 |
| | | | 7. Identification of emotional barriers to recovery and signposting to appropriate therapy, for example, GP/psychology | 57 | 4.7 |
| Neurological[35–38] | 39 (58%) | Clinical assessment | 1. Neural interface mobilisation | 98 | 8.1 |
| | | | 2. Functional neurological movement re-education | 7 | 0.6 |
| Movement restriction[39] | 59 (86%) | Clinical assessment | 1. Flexion mobilisation (grades 2–4) | 68 | 5.6 |
| | | | 2. Side flexion mobilisation (grades 2–4) | 5 | 0.4 |
| | | | 3. Extension mobilisation (grades 2–4) | 15 | 1.2 |
| | | | 4. Rotation mobilisation (grades 2–4) | 41 | 3.4 |
| | | | 5. Flexion+side flexion mobilisation (grades 2–4) | 11 | 0.9 |
| | | | 6. Flexion+side+flexion+rotation mobilisation (grades 2–4) | 62 | 5.2 |
| | | | 7. Extension+side flexion mobilisation (grades 2–4) | 0 | 0 |
| | | | 8. Manipulation (grade 5) | 0 | 0 |
| | | | 9. Seated mobilisation with movement (MWM) | 16 | 1.3 |
| | | | 10. Standing MWM | 16 | 1.3 |
| | | | 11. Mobilisation into functional position | 14 | 1.2 |
| | | | 12. Muscle stretches | 61 | 5.1 |
| | | | 13. Functional movement re-education | 7 | 0.6 |
| Understanding[40] | 66 (96%) | | 1. Management of erroneous believes relating to LRS provide education to help eradicate these beliefs | 57 | 4.7 |
| | | | 2. Pacing behaviours | 53 | 4.4 |
| | | | 3. Goal attainment | 58 | 4.8 |
| | | | 4. Health Promotion | 80 | 6.6 |
| | | | 5. Identification and treatment of central sensitisation-liaison with GP/pain clinic | 8 | 0.7 |
| | | | 6. Identification and treatment of peripheral sensitisation-liaison with GP/pain clinic | 7 | 0.6 |
| Conditioning[41 42] | 63 (91%) | Self-assessment answers, clinical interview and history | 1. Cardiovascular and conditioning exercise relevant to patients' goals | 83 | 6.9 |
| | | | 2. Function-specific stretches | 39 | 3.2 |
| | | | 3. Function-specific strengthening | 62 | 5.2 |
| | | | 4. Ergonomic advice | 14 | 1.2 |
| | | | 5. Ergonomic practice | 6 | 0.5 |
| | | | 6. Group exercise | 0 | 0.0 |
| | | | 7. Perturbation training | 7 | 0.6 |

**Table 3** Continued

| Domain | No of participants receiving component n=69 | Method of assessment | Treatment options | Frequency of component used | % |
|---|---|---|---|---|---|
| Movement control[43] | 33 (48%) | Clinical assessment | 1. Sagittal plane control in functional positions relevant to patients' problems/goals | 24 | 2.0 |
| | | | 2. Coronal plane control in functional positions relevant to patients' problems/goals | 15 | 1.2 |
| | | | 3. Axial plane control in functional positions relevant to patients' problems/goals | 1 | 0.1 |
| | | | 4. Multiplanar control in functional positions relevant to patients' problems/goals | 6 | 0.5 |
| | | | 5. Movement re-education in functional positions relevant to patients' problems/goals | 18 | 1.5 |
| Pain[44–46] | 52 (75%) | ODI VAS back and leg clinical interview and history | 1. Analgesic review and advice in liaison with GP/Pharmacist | 23 | 1.9 |
| | | | 2. Pain education | 60 | 5.0 |
| | | | 3. Pain coping strategies | 20 | 1.7 |
| | | | 4. Fear reduction intervention in liaison with psychologist/pain clinic | 12 | 1.0 |
| | | | 5. Stress reduction intervention in liaison with psychologist/pain clinic | 32 | 2.7 |
| Totals | | | | 1267 | 99.8%* |

*0.2% missing data, two treatment episodes where components not attributed.

GP, general practitioner; LRS, lumbar radicular syndrome; ODI, Oswestry Disability Index; VAS, Visual Analogue Scale.

## Acceptability of the primary and secondary outcome measures to patients and clinicians

The importance of examining acceptability of the outcome measures, processes and the intervention was a key area of investigation for the study, and the pilot trial included a qualitative element to explore these aspects. Details of the qualitative aspects of the study will be reported in forthcoming papers. However, in summary, the key processes necessary for implementation and evaluation of the study were reported to be acceptable by all stakeholders.

## Fidelity

Physiotherapists recorded the components of their treatment sessions at each patient encounter in order to enhance and measure treatment fidelity. Participants in the early intervention physiotherapy group had a mean of four treatment sessions and those participants in the usual care group three sessions. There were 269 physiotherapy sessions carried out as part of the POLAR study with 1267 component parts (table 3), 36 (3%) of which outside the protocolled treatment framework. The components outside the protocol consisted of three sessions of acupuncture and exercise other than that in the protocol. Video analysis was carried out independently on a purposive sample of five treatment sessions using a fidelity assessment tool developed by the lead author, clinical colleagues and PPIE (Patient and Public Involvement and Engagement) representatives. The maximum score for 'essential' aspects of fidelity was 15/15. The median score for the videos was 14/15 (93%) with a range of 13–15 (87%–100%).

## Sample size calculation for the definitive RCT trial

For the definitive RCT we propose the primary outcome is the ODI at 26 weeks postrandomisation as the ODI has shown to be acceptable to patients and a commonly used measurement of self-rated disability. In this pilot trial, we observed a difference in means (in favour of the control group) of 2.5 points (95% CI −4.5 to 9.1) between the randomised groups and an SD of 16 points at 26 weeks. There is a lack of consensus regarding the minimum clinically important difference for the ODI, with suggestions ranging from 6% to 30%.[30 31] Table 4 shows a range of sample sizes for varying target differences in the ODI. If we assume a target difference of five points on the ODI scale, then with 217 patients per group (434 in total) we would have 90% power to detect a five-point difference or more (equivalent to standardised effect size of 0.31) between the randomised groups which would be statistically significant at the 5% two-sided level. Allowing for a conservative estimate of 20% attrition (we observed 15% in this pilot), we would need to recruit and randomise 272 per group (544 in total).

Based on the recruitment rates observed in this trial of 80 patients in 8.5 months of recruitment at 10 centres (a rate of 0.9 patients per centre month), the main trial would need around 24 centres recruiting for 24 months to achieve this target.

**Table 4** Sample sizes for main randomised controlled trial for a range of target mean differences with a primary outcome of the Oswestry Disability Index score at 26 weeks postrandomisation

| Significance level (%) | Power (%) | SD | Target mean difference | Standardised effect size | No in each group | Total sample size (N) | Total sample size drop-out | |
|---|---|---|---|---|---|---|---|---|
| | | | | | | | 15% | 20% |
| 5 | 90 | 16 | 2 | 0.13 | 1346 | 2692 | 3168 | 3366 |
| 5 | 90 | 16 | 2.5 | 0.16 | 862 | 1724 | 2030 | 2156 |
| 5 | 90 | 16 | 3 | 0.19 | 599 | 1198 | 1410 | 1498 |
| 5 | 90 | 16 | 3.5 | 0.22 | 441 | 882 | 1038 | 1104 |
| 5 | 90 | 16 | 4 | 0.25 | 338 | 676 | 796 | 846 |
| 5 | 90 | 16 | 4.5 | 0.28 | 267 | 534 | 630 | 668 |
| 5 | 90 | 16 | 5 | 0.31 | 217 | 434 | 512 | 544 |
| 5 | 90 | 16 | 5.5 | 0.34 | 179 | 358 | 422 | 448 |
| 5 | 90 | 16 | 6 | 0.38 | 151 | 302 | 356 | 378 |
| 5 | 90 | 16 | 6.5 | 0.41 | 129 | 258 | 304 | 324 |
| 5 | 90 | 16 | 7 | 0.44 | 111 | 222 | 262 | 278 |
| 5 | 90 | 16 | 7.5 | 0.47 | 97 | 194 | 230 | 244 |
| 5 | 90 | 16 | 8 | 0.50 | 86 | 172 | 204 | 216 |
| 5 | 90 | 16 | 8.5 | 0.53 | 76 | 152 | 180 | 190 |
| 5 | 90 | 16 | 9 | 0.56 | 68 | 136 | 160 | 170 |
| 5 | 90 | 16 | 9.5 | 0.59 | 61 | 122 | 144 | 154 |
| 5 | 90 | 16 | 10 | 0.63 | 55 | 110 | 130 | 138 |

## DISCUSSION

This pilot study is the first to explore the role of early intervention physiotherapy for LRS. The study aimed to determine the feasibility of carrying out a full-scale RCT to determine the effectiveness of early physiotherapy for LRS. All of the feasibility parameters were found to be acceptable, including the set-up of GP centres to recruit participants, recruitment of participants and the retention of 85% of participants at 26 weeks. Both groups received the intervention at the appropriate time, within 2 weeks of randomisation for the early intervention physiotherapy group and after 6 weeks for the usual care group. The acceptance of the intervention, judged by the rate of attendance by participants at their treatment sessions, was better than anticipated.

There were some limitations to this study. First, although recruitment was satisfactory and ahead of time, the GPs involved in the study were well motivated and supportive of the study, in a city with a proven track record of GP involvement in service development and research. This may not be the case across the country and further afield. Similarly, the support of the service provider clinical, administrative and management staff was a key factor in the success of the study, a factor which may not be reproducible in other centres. Patients self-referred into the study after an introduction from their GP (a prerequisite for ethics approval) and so this group of patients may not be representative of a wider population. These factors need to be taken in account when planning a definitive study, and we have taken a more conservative view of attrition in the definitive sample size calculation. Our recommendations about recruitment also suggest including a wider geographical spread of GP centres to help meet the proposed recruitment rates. Site selection would need to consider current service provision and the ability to deliver the intervention in settings that are convenient and accessible to patients. The reliance on a clinical diagnosis of LRS made by the GP and physiotherapists is a potential limitation. The limitation being that there is likely to be a degree of diagnostic heterogeneity within the sample using a pathoanatomical model of care. There is, therefore, potential that participants with LRS in the study may have symptoms from something other than nerve root inflammation, including pseudoradicular symptoms, somatic or visceral referred symptoms.

The strengths of the study are that it was a pragmatic study in a clinical setting, using clinical staff and available resources and as such represents the real world of the NHS. We demonstrated that the study is feasible and the potential of early intervention physiotherapy to improve patient care.

## CONCLUSION

The POLAR study results indicate that a full-scale trial of early physiotherapy to treat patients with LRS is feasible. As there is a dearth of evidence about how and when best to treat this population, we conclude that a definitive trial is needed to help inform clinical practice.

**Acknowledgements** The authors would like to thank the members of the patient, public involvement and engagement (PPIE) group (Paul Colton, Amanda Dempsey, Rachel Harrison and Claire Village) for their invaluable support and guidance. Thank you to Jon Dickson, Mark Rhymes and Mark Pinkerton of the Yorkshire and Humber CRN. Thank you to the wonderful local physiotherapy service for their support, especially the fantastic physiotherapists who delivered the intervention and management (Sarah Withers and Helen Wilson). Finally and most importantly, a massive thank you to the participants.

**Contributors** MR instigated the idea for the study, developed the funding proposal and applied for funding. He developed the protocol, intervention handbook, gained ethical approval and acted as CI for the study. SJW is the primary supervisor for MRs' fellowship and contributed to the study conception, design and writing of the protocol and provided guidance with the statistical analysis. JC is an academic supervisor for MR and has provided specific guidance on protocol development, regulatory approvals and the design of the study. SKB provided input regarding the qualitative and mixed-method design and analysis aspects of the study. AC provided clinical supervision for MR. He has been involved in the conception of the study, its organisation, analysis and writing. All authors read and commented on drafts, and approved the final version of the manuscript.

**Funding** The lead author (MR) has received a personal Clinical Doctoral Research Fellowship (CDRF) award from Health Education England (HEE) and the National Institute of Health Research (NIHR). Award number: CDRF-2014-05-046.

**Disclaimer** This paper presents independent research funded by the National Institute for Health Research (NIHR). The views expressed are those of the author(s) and not necessarily those of the NHS, the NIHR or the Department of Health.

**Competing interests** None declared.

**Patient consent** Not required.

**Ethics approval** Ethical approval was received from NHS Scotland, East of Scotland Research Ethics Service (EoSRES) in August 2015 (REC reference 15/ES/0130). The study was conducted in accordance with the declaration of Helsinki and local governance requirements.

**Provenance and peer review** Not commissioned; externally peer reviewed.

**Data sharing statement** There are no additional data available for the study which has not been published. All data are available to anyone interested by contacting the corresponding author.

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
