## [Reviewer comments · BMJ Open]

ARTICLE DETAILS

TITLE (PROVISIONAL)	Does early intervention improve outcomes in the physiotherapy management of Lumbar Radicular Syndrome? Results of the POLAR Pilot Randomised Controlled Trial
AUTHORS	Reddington, Michael; Walters, Stephen; Cohen, Judith; Baxter, Susan; Cole, Ashley

VERSION 1 – REVIEW

REVIEWER	Steven P Cohen Johns Hopkins, USA
REVIEW RETURNED	24-Jan-2018

GENERAL COMMENTS	The authors have performed a small randomized, exploratory study designed to determine whether early PT provides more benefit than a standard PT regimen. I think this will be an important, ambitious study, and I applaud the authors for undertaking this (and look forward to reading the manuscript that results from completing the study). My main concern is basically that with required clinical trials registration, "reporting" a study in a peer-reviewed journal has little purpose and is unlikely to be cited. 1. Abstract: Define LRS. Also, since results have been obtained, why are they not reported.2. In general, the introduction requires better referencing and there are several grammatical errors. For example, studies generally do not show long-term benefit from surgery compared to non-surgical treatment.3. In the introduction, the authors cite a randomized trial that shows that early PT may be beneficial; yet, they do not state how their study will add to the literature (i.e. how is it different?). There is also no hypothesis listed.4. Please clarify and provide a reference for the levels of disability (i.e. I believe most use 20% as the cutoff for mild and moderate disability). Also, if "mild and moderate are lumped together, then what is purpose of writing "< 22-40%")? Why not just write < 40%?5. What were the selection criteria? It's not clear how a diagnosis of radicular pain was made (e.g. MRI findings, physician determination, solely reported symptoms, neurological signs), and since mechanical, non-neuropathic pain often radiates to the leg, this needs to be noted.6. Isn't "up to" 6 sessions on the low side, as we often refer patients for twice or three times per week x 8-12 weeks (some insurance companies requiring 3 months of PT failure before authorizing interventions). This speaks to generalizability.7. Page 7, para 1: Do the authors mean "PROMIS" (rather than PROMS)?8. Page 8: Please specifically note how many patients refused
---

	participation? 9. For the MCID for ODI, consider using the findings by Copay et al. 2008 (12.8%). 10. On a related note, please consider defining what a “responder” is (i.e. positive outcome). 11. How will the authors deal with co-interventions (e.g. medications, integrative treatments and procedural interventions), as these can affect their primary outcome?
--	--

REVIEWER	Dr. Andrew Hahne La Trobe University, Melbourne, Australia
REVIEW RETURNED	01-Feb-2018

GENERAL COMMENTS	This is a thorough report of pilot study results for a prospective RCT in a field that does require more trials to be undertaken. I have some areas of concern, some of which are too late to correct for the current pilot trial, but it may not be too late to consider prior to launching the definitive trial that the authors propose to undertake. In any study relating to LRS or “sciatica”, it is always a concern that a highly heterogenous population will end up in the study if the definition and selection criteria for the condition under investigation are not tightly defined. In this study (based on my reading of this paper and the trial protocol published earlier) the diagnosis of LRS appears to have been left up to numerous GP’s in their clinical practices without any recommendations as to what will constitute acceptable evidence of the diagnosis for the purposes of the study. It is unlikely that all of the GP’s would share the same opinion as to what constitutes a valid diagnosis of LRS, and research shows that there are numerous opinions as to what criteria need to be met to be confident that a person has LRS (eg. Genevay S, Atlas SJ, et al. Variation in Eligibility Criteria From Studies of Radiculopathy due to a Herniated Disc and of Neurogenic Claudication due to Lumbar Spinal Stenosis: A Structured Literature Review. Spine (Phila Pa 1976). 2010;35(7):803-11.). Some GP’s may only diagnose LRS if a neurological sign is present and imaging shows nerve root compression (ie. quite a definitive diagnosis) while others may make the diagnosis based on pain distribution alone (ie. a highly questionable diagnosis). It may be too late for the pilot study, but I would recommend that this issue is given careful consideration before a definitive trial is commenced to be sure that all of the involved GP’s are using similar and convincing diagnostic criteria. Without this, the pilot trial and the eventual definitive trial will contain a highly heterogenous sample that will include some people with LRS, and others with simple referred pain from their back (non-radicular pain), others with alternative musculoskeletal problems (hip pain referred into the leg, hamstring strains), and others with non-musculoskeletal pain (peripheral neuropathy or peripheral vascular disease / vascular claudication). I question the authors’ decision to remove the upper age limit for inclusion in the definitive trial of 70 years. This will increase the likelihood of including people without LRS (eg. increased rate of peripheral vascular disease and peripheral neuropathy), and will also increase the rate of spinal stenosis V’s disc herniations (and these two conditions may need to be treated differently). The authors plan to use the 26-week Oswestry outcome as the primary endpoint in the main trial. From my perspective this is a
---

questionable decision. The whole premise of this study is to see if earlier intervention with physiotherapy is effective, partly so that it can be quickly determined whether surgery will be needed in people with LRS before they move outside of the “window of opportunity” where surgery may be most helpful. The critical priority then is to determine as quickly as possible whether someone is going to respond to physiotherapy, hence the 6-week and 12-week outcome timepoints would appear to be the most critical. The aim of the study (based on the rationale presented by the authors) could be fully achieved if early physiotherapy were to result in a faster improvement at 6-weeks, even if the results of the groups were similar at 26-weeks, because by improving faster (at 6 weeks or 12 weeks) then surgery is more likely to be avoided – the 26-week outcome then becomes more of a secondary consideration given the main aim has already been achieved. Ultimately it is your study and your decision, but my argument above seems more consistent with the premise and aim of the study.

I have some concern over the sample size estimation for the definitive trial. The trial will be powered to find a difference of 5 points on the Oswestry at 26-weeks. However, in the pilot study (with quite a sizable sample of 80 participants), the estimated effect at that timepoint was only 2.5 points (and in fact it favoured the usual care group over the early physiotherapy group). What makes you confident that a definitive trial would be likely to lead to a larger effect size to the one you obtained in your relatively large pilot trial? I also note with interest that at 6 weeks the effect size appeared to be approximately 5 points in favour of early physiotherapy, providing further support to my previous point about using the 6-week outcome timepoint as the primary endpoint in the definitive trial.

Did any participants undergo surgery during the follow-up period? If there were none, it would be worth reporting that as it is obviously of great interest.

The longer symptom duration at entry to the study in the early physiotherapy group is highly problematic. The duration of pain in the early physiotherapy group was 33% higher than the usual care group. In fact, if you looked at the time to treatment commencement from onset of symptoms (rather than time to treatment from consulting your GP), then the “early physiotherapy” group commenced treatment an average of 103 days after onset of symptoms, compared to the usual care group commencing treatment an average of 104 days after symptom onset. So the study is only testing whether starting physiotherapy one day earlier in your course of LRS makes a difference? This is so problematic that the authors need to ensure that this is not repeated in the definitive trial. I would recommend stratifying the randomisation for symptom duration (I think this is a more important stratification variable than Oswestry Score) to ensure that the duration of symptoms ends up similar in both groups. Only then can you fairly compare whether “early physiotherapy” is beneficial or not. You could also consider excluding people with very long symptom durations when they present to their GP, because again it becomes incorrect to suggest that someone with a symptom duration of (for example) 12 months is receiving “early physiotherapy” at all because it is already very late in their course.

I wonder if you would consider commenting on the difference in EuroQOL total utility score at baseline between the groups? This

	appears to me to be a sizable difference at baseline (and it is unusual to see such a large discrepancy in this particular outcome measure between groups at any stage in a study, let alone at baseline). Table 2: I wondered if it may be more (or additionally) helpful to report the proportion of participants (rather than the proportion of treatment sessions) who received each of the intervention components. Some of the listed components may be important for each patient to receive at some time in their course of treatment, but may not necessarily be important to deliver at every treatment session that they attend. It would therefore be useful to see how many participants received each component at some stage in their treatment. Minor additional points: Abstract - The abbreviation “LRS” appears on its first instance in the Abstract. Please spell this term out in full, at least for its first appearance in the abstract. Abstract - Under the “intervention: subheading it is not at all apparent what the difference is between the two groups in this RCT. We are only told that both groups received physiotherapy, and it does not mention in this section that one group started their physiotherapy early while the other group started later. This is definitely worth clarifying under the intervention heading.
--	---

VERSION 1 – AUTHOR RESPONSE

Feedback from reviewer	Author response
Editors reply	
Please remove the 'ethics and dissemination' section from the abstract	Thank-you, I have corrected the section.
Response to Dr Cohens’ review and advice	
Abstract: Define LRS. Also, since results have been obtained, why are they not reported.	I have defined LRS in the abstract and presented the ODI results.
In general, the introduction requires better referencing and there are several grammatical errors. For example, studies generally do not show long-term benefit from surgery compared to non-surgical treatment.	I have made amendments to the body of the introduction, in particular regarding the long-term outcomes of surgery and conservative management and enhanced the references appropriately. I have also attempted to improve the syntax and grammar in the introduction to improve its reading.
In the introduction, the authors cite a randomized trial that shows that early PT may be beneficial; yet, they do not state how their	The cited RCT was for LBP, not LRS and as such serves to show the lack of research in this regard.

study will add to the literature (i.e. how is it different?). There is also no hypothesis listed.	I have therefore stated this in the text.
Please clarify and provide a reference for the levels of disability (i.e. I believe most use 20% as the cutoff for mild and moderate disability). Also, if "mild and moderate are lumped together, then what is purpose of writing "< 22-40%")? Why not just write < 40%?	The levels of disability have been taken and aggregated according to the original disability levels described by Fairbank et al 1980. The ODI levels could have been described as <40% of but I thought it preferable to delineate the <22% disability level as this is considered to be a 'normal' level of 'disability'.
What were the selection criteria? It's not clear how a diagnosis of radicular pain was made (e.g. MRI findings, physician determination, solely reported symptoms, neurological signs), and since mechanical, non-neuropathic pain often radiates to the leg, this needs to be noted.	Participant selection was made clinically by G.Ps who had received training by the author in identifying patients with LRS. I agree this method is not infallible but reflects clinical practice in general practice in the U.K and thus enhances clinical credibility and generalisability in primary care.
Isn't "up to" 6 sessions on the low side, as we often refer patients for twice or three times per week x 8-12 weeks (some insurance companies requiring 3 months of PT failure before authorizing interventions). This speaks to generalizability.	I agree, it is certainly on the low side, however this exemplifies a significant difference in healthcare systems across the world. The NHS service providers could not countenance 36 treatment sessions for a patient with LRS due to the cost. In primary care the mean number of treatment sessions for a patient with LRS is 2.8. Further pressure and research is required to rectify this.
Page 7, para 1: Do the authors mean "PROMIS" (rather than PROMS)?	The term PROMS refers to Patient Reported Outcome Measures and not the PROMIS healthcare measures system used in the USA.
Page 8: Please specifically note how many patients refused participation?	2 participants 'refused' to participate or enter the study. One found the travel to the treatment site onerous, the other was unhappy with the group allocation post randomisation.
For the MCID for ODI, consider using the findings by Copay et al. 2008 (12.8%).	Thank-you for the useful reference. As you are aware there are a variety of estimates for the MCID for the ODI quoted in the literature ranging from 6% to 30% and most points between. We have been conservative and used the lower value for the target difference in the sample size calculation for the main trial.
On a related note, please consider defining what a "responder" is (i.e. positive outcome).	It is difficult to define a 'responder' in terms of this study as it was not a full-scale trial investigating the effectiveness of treatment and as aforementioned the MCID for ODI has such a

	large range of options. Furthermore the difficulties in dichotomising continuous data have been presented by Altman & Royston (among others) BMJ stats notes 2006; 332; 1080).
How will the authors deal with co-interventions (e.g. medications, integrative treatments and procedural interventions), as these can affect their primary outcome?	Participant data was collected, but not reported in this paper for other co-interventions the participants utilised such as analgesia/other treatments (there were no co-treatments reported). This is a pragmatic trial (not explanatory), designed to compare the policy of early intervention vs usual care. No change was therefore made to the analysis, in any future trial these will be part of any analysis a priori.
Response to Dr Hahnes' review and advice	
Diagnosis of LRS and the heterogeneous nature of the LRS population.	Each G.P participant received (around) 30 minutes of training in the study processes and patient identification, specifically the type of pain LRS sufferers present with, dermatomal areas and clinical tests (neurological testing including power, sensation, reflexes and provocation tests (SLR, Slump)). The diagnosis was also checked by the main author when gaining consent from the participant and further verified (or not) by the treating physiotherapist on initial examination. Whilst a clinical diagnosis of LRS is certainly not infallible, it does represent the clinical reality in the U.K healthcare system where an MRI scan for each patient presenting with radicular-type symptoms is not financially feasible, nor clinically necessary. For the definitive trial I am considering using MRI as a validation tool for clinical diagnosis.
I question the authors' decision to remove the upper age limit for inclusion in the definitive trial of 70 years. This will increase the likelihood of including people without LRS (eg. increased rate of peripheral vascular disease and peripheral neuropathy) and will also increase the rate of spinal stenosis V's disc herniations (and these two conditions may need to be treated differently).	This is a valid point, the argument for which was the starting point in the study, before submitting a protocol amendment to remove the age limit. The rate of other co-morbidities mimicking LRS certainly increases with age, however treatment was dependent on the patients' individual needs and requirements rather than the technical/ medical/structural diagnosis as to whether their symptoms were caused by a disc protrusion or bony stenosis or a combination of both. In clinical practice we regularly see radiculopathy (rather than the anticipated stenotic symptoms) caused by a primary disc protrusion in the elderly (over

	65s), thus the age limit was removed.
The authors plan to use the 26-week Oswestry outcome as the primary endpoint in the main trial. The critical priority then is to determine as quickly as possible whether someone is going to respond to physiotherapy, hence the 6-week and 12-week outcome time points would appear to be the most critical	Thank-you for highlighting this very valid point, we are interested in both the short and longer-term outcomes; which the AUC outcome which we report captures. I note the importance of presenting the longer-term effects in terms of recurrent healthcare usage (to make an argument for cost-savings) as well as the longer-term effects on the individuals function, return to work/activities.
I have some concern over the sample size estimation for the definitive trial. I also note with interest that at 6 weeks the effect size appeared to be approximately 5 points in favour of early physiotherapy, providing further support to my previous point about using the 6-week outcome time-point as the primary endpoint in the definitive trial.	The sample size calculation is valid for detecting a target difference of five points at 6 weeks or 12 weeks or 26 weeks assuming the standard deviation of the outcome (The ODI) is around 16 points at each of these three-time points. In actual fact the variability of the ODI at 6 weeks – see Table 3 – was 16.1 and 18.7 points in the intervention and group groups respectively compared to 11.3 and 15.5 at 26 weeks. Therefore, based on this we would require a larger sample size for a study with a primary endpoint at 6 weeks post randomisation.
Did any participants undergo surgery during the follow-up period? If there were none, it would be worth reporting that as it is obviously of great interest.	I have noted the 2 participants who underwent surgery in the research results.
The longer symptom duration at entry to the study in the early physiotherapy group is highly problematic.	Thank-you for pointing this out, the results illustrate a large Standard Deviation in the duration of symptoms, particularly in the early intervention physiotherapy group with 1 participant having had duration of 1796 days (He sought treatment only when the symptoms became unbearable). We have now reported the median, rather than mean due to the skewed distribution and IQR as being more appropriate due to the outliers. All participants were randomised and any differences are likely to be due to chance.
I wonder if you would consider commenting on the difference in EuroQOL total utility score at baseline between the groups? This appears to me to be a sizable difference at baseline (and it is unusual to see such a large discrepancy in this particular outcome measure between groups at any stage in a study, let alone at	Yes potentially; Walters and Brazier suggest the MCID for the EQ-5D is around 0.07 points – so this is bigger than that. Since the patients were randomised any differences should just be due to chance; and the patients were well matched on most of the other

baseline).	baseline variables. Walters S.J., & Brazier J.E. Comparison of the minimally important difference for two health state utility measures: EQ-5D and SF-6D. Quality of Life Research, 2005; 14:1523-1532.
Table 2: I wondered if it may be more (or additionally) helpful to report the proportion of participants (rather than the proportion of treatment sessions) who received each of the intervention components.	The number of participants receiving each of the intervention components has been added to table 2.
Abstract - The abbreviation "LRS" appears on its first instance in the Abstract. Please spell this term out in full, at least for its first appearance in the abstract.	The LRS abbreviation has been amended in the abstract.
Abstract - Under the "intervention: subheading it is not at all apparent what the difference is between the two groups in this RCT. We are only told that both groups received physiotherapy, and it does not mention in this section that one group started their physiotherapy early while the other group started later. This is definitely worth clarifying under the intervention heading.	The treatment allocation and differences between the groups has been amended in the abstract.

VERSION 2 – REVIEW

REVIEWER	Dr. Andrew Hahne La Trobe University, Australia
REVIEW RETURNED	19-Mar-2018

GENERAL COMMENTS	The majority of my points have been addressed by the authors, there are just a few outstanding issues. Major comments: Participants: The authors have clarified in the response to my comments that there was at least an attempt to create some consistency in the method that was used to diagnose LRS. The training of GP's (albeit only for 30 minutes), cross-checks by the researchers, and further cross-checks by the treating physiotherapists. However, your trial flow-chart shows that nobody was excluded based on the cross-checks by the researchers or the physiotherapists, which makes me question how thoroughly anyone was assessing the participants (did three people all really agree on a diagnosis of LRS in every single case that presented to the trial – the literature would not support
--

	such a strong level of diagnostic agreement for this condition). I wonder if you would consider, for readers of the final paper, the following two further concessions: 1) You have stated that “Each participating G.P underwent training and were equipped with a diagnostic aide memoire for clinically identifying patients with LRS”. It would seem appropriate, given the high level of disagreement in the literature about what constitutes a diagnosis of LRS, to publish your diagnostic aide that was given to GPs, perhaps as an appendix, so that readers can at least see what diagnostic criteria were aspired to. 2) I do feel that a couple of sentences could be added to the discussion relating to the high likelihood of diagnostic heterogeneity in your sample. The fact that you are considering using MRI in your future study suggests that you agree it is somewhat of a problem, so why not acknowledge this to readers in the discussion? Sample size estimation for the definitive study: I accept the counter-argument provided about setting the primary timepoint for the definitive study. However, I am still questioning why the definitive study should proceed based on the effect sizes obtained in the pilot study? You are aiming for a between-group difference of 5 points on the Oswestry at 26 weeks (and are powering your trial accordingly), but the pilot study shows a between-group difference at this timepoint of -2.5 points (favouring the usual care group)! What makes you think that a definitive trial will lead to a larger effect size than what you could obtain in the pilot study? Pilot studies of this size are not just for determining feasibility and standard deviations, they also give a reasonably good indication as to what effect size you could expect from a larger study. On the basis of the pilot study results, I would not recommend repeating an identical study on a larger population and expecting a different result. So my question is, what will be changed in the definitive study about the selection criteria / recruitment / diagnosis / treatment methods etc so that they are not simply “feasible” but will also provide the best possible chance of reaching the target effect size of 5 points? Minor points: Table 2: Thank-you for adding to this table the proportion of participants who received each intervention. However, the percentage is missing from the “understanding” treatment domain (the percentage was listed for all the other treatment domains other than this one), if you have this percentage could it be added too? Table 3 – can you please annotate in the table footnote whether a negative value for the between-group AUC represents a better outcome in the early physiotherapy or usual care group as this is currently unclear. Also, is the Back Pain AUC between-group difference correct, because it does not seem to match the AUC values for each group, and I wonder if the sign should be negative? The AUC result mentioned in the abstract needs clarifying too, as again it is not clear which group has done “better” from the way these data are presented.
--	--

VERSION 2 – AUTHOR RESPONSE

Feedback from reviewer	Author response
------------------------

Editors reply	
Reference numbering errors	Thank-you for highlighting these errors. They have been duly corrected.
The manuscript still contains grammatical errors, which is affecting the readability of your work. Please work to improve the quality of the English throughout your manuscript.	Thank-you for highlighting the grammatical errors in the submitted paper. I have corrected these throughout.
We are concerned about your response to the following question from reviewer 1: “What were the selection criteria?” Can you elaborate on what this training consisted of? How were patients identified? How does this training reflect clinical practice in general practice in the U.K? You note that this issue is a limitation in your response to reviewer 1 but it is not discussed as a limitation in the paper.	The selection criteria for and identification of participants is outlined on page 6. The training G.Ps underwent is detailed in supplementary material. This additional training for G.Ps is a revision for Sheffield G.Ps who undergo this training as part of their specialist training. I am unable to comment about national G.P training.
Response to Dr Hahnes’ review and advice	
I wonder if you would consider, for readers of the final paper, the following two further concessions: 1) You have stated that “Each participating G.P underwent training and were equipped with a diagnostic aide memoire for clinically identifying patients with LRS”. It would seem appropriate, given the high level of disagreement in the literature about what constitutes a diagnosis of LRS, to publish your diagnostic aide that was given to GPs, perhaps as an appendix, so that readers can at least see what diagnostic criteria were aspired to. 2)I do feel that a couple of sentences could be added to the discussion relating to the high likelihood of diagnostic heterogeneity in your sample. The fact that you are considering using MRI in your future study suggests that you agree it is somewhat of a problem, so why not acknowledge this to readers in the discussion?	 1. Thank-you for outlining what is a contentious issue. I acknowledge the difficulty of clinically diagnosing a patho-anatomical structure (? a disc prolapse in this instance) due to the lack of validity/specificity of the clinical tests and the heterogeneity of the study population. However, this was not the intention of the study. The intention was to establish whether the participant had a group of symptoms indicative of LRS, suggesting lumbar nerve root inflammation. It does in no way propose that this is due to a specific structure, disc or otherwise. The treatment is aimed at the person and their functional problems rather than at ameliorating a patho-anatomical structure. 2. The suggested use of MRI for the future study will help in this regard, but this simply wasn't feasible in the pilot study due to costs. I have added an explanation of this limitation to the discussion.
Sample size estimation for the definitive study: So my question is, what will be changed in the definitive study about the selection criteria /	We have included a new table with sample size for a range of target mean differences, in the ODI outcome, from 2 to 10 points. In this pilot trial, we

recruitment / diagnosis / treatment methods etc so that they are not simply “feasible” but will also provide the best possible chance of reaching the target effect size of 5 points?

observed a difference in means (in favour of the control group) of 2.5 points (95% CU: -4.5 to 9.1) between the randomised groups and a standard deviation of 16-points at 26 weeks. As the confidence interval for the treatment difference includes zero and differences of 2.5 points, then the results of the pilot study could be considered to be equivocal. There could be no difference between treatments, or there could be a difference larger than the 2.5 points; the results would not preclude either possibility.

This approach is superior to formal hypothesis testing as there is insufficient power to test hypotheses, and its focus on the MCID will help inform the main confirmatory trial. Several authors have cautioned against using the estimated effect (the observed mean difference) or the effect size from the pilot study. (Kraemer et al 2006; Moore et al 2011). If the estimate was large in the pilot study, using it would result in largely underpowered trials. Also, the decision to move forward to a larger trial should not solely be based on the estimated effect size from the pilot study. If the estimate was small in the pilot study, using it would result in a high likelihood of not investigating truly efficacious interventions. Because pilot clinical trials are usually small, they have a high likelihood of imbalances at baseline, leading to unreliable estimates of treatment effects.

Kraemer HC1, Mintz J, Noda A, Tinklenberg J, Yesavage JA.

Caution regarding the use of pilot studies to guide power calculations for study proposals. Arch Gen Psychiatry. 2006 May;63(5):484-9.

Andrew C. Leon, Ph.D.,¹ Lori L. Davis, M.D.,² and Helena C. Kraemer, Ph.D.³ 2012 The Role and Interpretation of Pilot Studies in Clinical Research. Psychiatr Res. 2011 May; 45(5): 626–629.

Charity G. Moore, Ph.D.,¹ Rickey E. Carter, Ph.D.,² Paul J. Nietert,³ and Paul W. Stewart, Ph.D.⁴

Recommendations for Planning Pilot Studies in

	Clinical and Translational Research. Clin Transl Sci. 2011 Oct; 4(5): 332–337.
Table 2: Thank-you for adding to this table the proportion of participants who received each intervention. However, the percentage is missing from the “understanding” treatment domain (the percentage was listed for all the other treatment domains other than this one), if you have this percentage could it be added too?	Thank-you for highlighting this. Amended.
Table 3 – can you please annotate in the table footnote whether a negative value for the between-group AUC represents a better outcome in the early physiotherapy or usual care group as this is currently unclear. Also, is the Back Pain AUC between-group difference correct, because it does not seem to match the AUC values for each group, and I wonder if the sign should be negative? The AUC result mentioned in the abstract needs clarifying too, as again it is not clear which group has done "better" from the way these data are presented.	Added footnotes to Table 3; and additional text to the abstract.

VERSION 3 – REVIEW

REVIEWER	Dr. Andrew Hahne La Trobe University, Australia
REVIEW RETURNED	14-May-2018

GENERAL COMMENTS	The revised manuscript has satisfactorily addressed my outstanding concerns from the last version. I just have one more minor suggestion that could easily be addressed - On page 3 under strengths and limitations one of the dot points says that "The diagnosis of LRS was made from the clinical history and examination.....". However, nowhere else in the paper does it appear to state anything about physical examination, and the diagnostic aid provided to medical practitioners also does not require examination - the diagnosis of LRS was based on the nature and location of symptoms only - not physical examination. It would therefore seem more accurate to re-word the strengths / limitations to "The diagnosis of LRS was made based solely on clinical symptoms...." If physical examination was required to confirm the diagnosis, it would seem important to explain this in the body of the paper.
--

VERSION 3 – AUTHOR RESPONSE

Feedback from reviewer	Author response
Response to Dr Hahnes' review and advice	
I just have one more minor suggestion that could easily be addressed - On page 3 under strengths and limitations one of the dot points says that "The diagnosis of LRS was made from the clinical history and examination.....". However, nowhere else in the paper does it appear to state anything about physical examination, and the diagnostic aid provided to medical practitioners also does not require examination - the diagnosis of LRS was based on the nature and location of symptoms only - not physical examination. It would therefore seem more accurate to re-word the strengths / limitations to "The diagnosis of LRS was made based solely on clinical symptoms...." If physical examination was required to confirm the diagnosis, it would seem important to explain this in the body of the paper.	Thank-you for your helpful response, changes have been made to accommodate the suggestion in the strengths and limitations.